# Cation Charge as a Tool to Change Dimensionality in Organic–Inorganic Hybrids Based on Copper Thiocyanate Templated by 1,4-Diazabicyclo[2.2.2]octane

**DOI:** 10.3390/molecules28083608

**Published:** 2023-04-20

**Authors:** Evgeny Goreshnik, Svitlana Petrusenko

**Affiliations:** Department of Inorganic Chemistry and Technology, Jožef Stefan Institute, Jamova 39, 1000 Ljubljana, Slovenia

**Keywords:** dabco, thiocyanate, copper, coordination polymer

## Abstract

The first three compounds based on a {copper–thiocyanate–dabco} combination, namely, (Hdabco)[Cu_2_(NCS)_3_] (**1**), (H_2_dabco)[Cu(NCS)_3_] (**2**), and [Cu(Hdabco)_2_(NCS)_4_]∙2dmso (**3**), where dabco = 1,4-diazabicyclo[2.2.2]octane were synthesized and characterized by single-crystal XRD, elemental analysis, Raman, and partial IR spectroscopy. In copper(I) derivatives, the influence of the charge of the organic cation on the dimensionality of the crystal structure is observed. Thus, in the case of **1**, monoprotonated Hdabco^+^ cations provide the template for the formation of a polymeric anionic 3D framework {[Cu_2_(NCS)_3_]^−^}_n_, while in the case of **2**, diprotonated H_2_dabco^2+^ cations together with discrete [Cu(SCN)_3_]^2−^ anions generate a simple ionic 0D structure with an island-like crystal lattice. The anionic {[Cu_2_(SCN)_3_]^−^}_n_ framework has infinite square channels of 10 × 10 Å size running along the 001 crystallographic direction. In **3**, both the Hdabco^+^ and thiocyanato units behave as terminal monodentate ligands attached to copper(II) centers via N-donor atoms, forming neutral molecular complexes with an elongated (4+2) octahedral environment. The crystallization molecules of dmso are hydrogen bonded to the protonated parts of the coordinated dabco molecules. A series of by-products Cu(SCN)_2_(dmso)_2_ (**4**), (Hdabco)SCN (**5**), (H_2_dabco)(SCN)_2_ (**6**), and (H_2_dabco)(SCN)_2_∙H_2_O (**7**) were identified and characterized.

## 1. Introduction

Dabco (1,4-diazabicyclo[2.2.2]octane) is one of the most popular building blocks in crystal engineering and materials chemistry mainly due to the following features: (a) specific globular symmetry; (b) ability of sequential protonation to form cations with +1 or +2 charges; (c) typical bidentate bridging coordination mode; (d) availability of two donor sites for hydrogen or other noncovalent bonding; (e) thermal and redox stability; (f) catalytic properties. The recent discovery of strong ferroelectric properties for (H_2_dabco)(NH_4_)Cl_3_ provided new impetus to research in this area [1,2]. The CSD search (ver. 2022.3.0) reveals more than 2000 hits for compounds containing dabco or its derivatives. Unexpectedly, the search for structures containing both dabco and thiocyanate at the same time proved to be rather limited: 7, 7, and 2 hits for Hdabco, dabco, and H_2_dabco, respectively. No data were found for copper thiocyanate complexes with dabco. Copper(I) thiocyanate itself has attracted attention as a p-type semiconductor for solar applications. In addition, copper(I) thiocyanate, as well as many of its derivatives, is a known luminescent compound [3]. Recently, luminescent thiocyanate-based coordination polymers with intriguing properties have been reported [4]. 3d^10^-metal thiocyanate-based coordination compounds could form photoluminescent coordination polymer glasses [5,6]. Considering possible synergistic effects of the different components in one compound, the combinations {dabco–thiocyanate} and {dabco–thiocyanate–copper} seem to be worth studying not only from a theoretical point of view but also because of their potential applicability.

Recently, one of the authors successfully used protonated dabco derivatives as templating agents for the preparation of copper-chloride coordination polymers with various dimensionality [7]. Different binding modes of the thiocyanate-anion in combination with tetrahedrally coordinated Cu^+^ ions appear very promising for the formation of three-dimensional coordination polymers.

Based on the above considerations, we decided to synthesize copper thiocyanate derivatives templated by protonated dabco. Here we report the syntheses and crystal structures of three copper thiocyanate-based complexes (Hdabco)[Cu_2_(NCS)_3_] (**1**), (H_2_dabco)[Cu(NCS)_3_] (**2**), and [Cu(Hdabco)_2_(NCS)_4_]∙2dmso (**3**). Metal-free dabco thiocyanates (Hdabco)NCS (**5**), (H_2_dabco)(NCS)_2_ (**6**), and (H_2_dabco)(NCS)_2_∙H_2_O (**7**) and a new complex [Cu(NCS)_2_(dmso)_2_] (**4**) were also isolated and characterized as side products.

## 2. Results and Discussion

### 2.1. Synthesis

Complexes **1** and **3** were obtained via the “ammonium salt direct synthesis” approach [8] using unactivated copper metal powder, ammonium thiocyanate, and dabco in dmf/dmso (1:1) solution (Figure 1).

High temperature, long reaction time, and an additional excess of ammonium thiocyanate contribute to the formation of **3**, while the opposite conditions are preferable for **1**. It should be noted that during long-term storage of **1** in the mother solution, the colorless prismatic crystals of **1** gradually transform into red needle-like crystals of the known compound Cu_3_(NCS)_4_(dmso)_2_ [9], which later transform into light green lamellar crystals of compound **4**.

The use of H_2_SO_4_ or HCl for acidification leads to the formation of the more stable products (H_2_dabco)SO_4_H_2_O [10] or (H_2_dabco)(NH_4_)Cl_3_ [1]. CuSCN and CuCN were tried as a source of copper(I) for the synthesis of **1,** but in all cases, the target compound was obtained as a mixture with various products, among which compounds **2** and **5**–**7** were identified (Figure 2). For compounds **5** and **6,** preparation methods were elaborated.

The formation of compounds **1**–**4** can be understood if one considers the following reaction schemes:2Cu + 3NH_4_NCS + ½O_2_ + dabco → (Hdabco)[Cu_2_(NCS)_3_] + 3NH_3_ +H_2_O
Cu + 4NH_4_NCS + ½O_2_ + 2dabco + 2dmso → [Cu(Hdabco)_2_(NCS)_4_]∙2dmso + 4NH_3_ +H_2_O
Cu_3_(NCS)_4_(dmso)_2_ + 2NH_4_NCS + ½O_2_ + 4dmso → 3Cu_2_(NCS)_2_(dmso)_2_ + 2NH_3_ +H_2_O

### 2.2. Crystal Structures

A summary of the crystallographic data and structure refinement are given in Table 1. The crystallographically independent part of structure **1** contains two metal centers, a mono-protonated Hdabco^+^ cation, and three thiocyanate anions (Figure 1). The Cu1 ion has a tetrahedral environment consisting of one nitrogen atom and three sulfur atoms, while the Cu2 center is surrounded by two N and two S atoms. The S1 atom is bound to only one metal ion, while the S2 and S3 atoms act as a μ_2_ bridge between two copper centers. The Cu–S distances vary between 2.330(4) and 2.432(4) Å, and the Cu–N bond lengths are 1.946(13)–1.966(13) Å. The shortest Cu–S distance corresponds to the lowest steric hindrance in the case of the S1 atom bound to only one metal ion. Owing to the bridging role of the N2C2S2^-^ and N3C3S3^-^ moieties, the copper ions are linked into tetrameric columns running along the 001 crystallographic direction. The S1C1N1^-^ units are responsible for linking the aforementioned columns into a three-dimensional network of 3,5T8 topology with infinite channels of approximately 10 × 10 Å size (Figure 2). A similar 3D framework was found in the structure of [(CH_3_)_4_N]Cu_2_(SCN)_3_ [11]. The Hdabco monocations, located inside these channels in head-to-tail mode, are interconnected by N-H…N hydrogen bonds (Appendix A) to form infinite chains. The Hdabco moiety exhibits a moderately twisted conformation with a range of N–C–C–N torsion angles from 12.8° to 16.2°. The geometry around protonated and non-protonated nitrogen atoms in the Hdabco^+^ cation differs slightly: the N4–C bond lengths are 1.46(2)–1.48(2) Å, and the N5–C distances are 1.49(2)–1.50(2) Å.

The structure of compound **2** consists of discrete H_2_dabco^2+^ cations and [Cu(SCN)_3_]^2−^ anions (Figure 3). Three symmetrically related sulfur atoms of three SCN^−^ anions form a trigonal-planar coordination surrounding the copper center located on the 
$\bar{6}$
 axis. To the best of our knowledge, this is the first known example of such an anion. The Cu–S distances of 3 × 2.2304(7) Å appear, based on the CCDC database, to be the shortest known distances Cu–SCN. The H_2_dabco^2+^ moieties, which lie on three axes passing through both N atoms, have an ideally untwisted conformation. The C-C and N-C distances are in the same range as observed for other H_2_dabco^2+^ salts [7]. Cations and anions are connected via weak N-H...N(SCN) hydrogen bonds (Appendix A).

An island-type structure of compound **3** consists of Cu(SCN)_4_^2−^ moieties with two attached via N–Cu bonds two Hdabco^+^ ligands. Each [Cu(Hdabco)_2_(SCN)_4_] complex unit is bound to two dmso molecules via N-H…O hydrogen bond (Figure 4). Due to the higher oxidation state of the central atom, the thiocyanate moieties are bound to the Cu^2+^ center via the N atoms. The Cu–N(SCN) distances are almost the same (1.9842(16)–1.9953(16) Å) and agree well with the CCDC 1.93–2.03 Å range. The two apical Cu–N(Hdabco) distances are strongly elongated to 2.05 Å. The Hdabco^+^ geometry is very similar to that of compound **1**. The main difference is the almost untwisted conformation (the N–C–C–N angles are 4.3–6°). There are two types of weak interactions responsible for the assembly of [Cu(Hdabco)_2_(SCN)_4_] and dmso moieties. The O atom of the dmso molecule is bound to the Hdabco unit via N4-H4...O1 hydrogen bond (Appendix A). Additionally, weak S(dmso)…S(SCN) contacts of 3.410 Å are present in structure **3**.

In the crystal structure of compound **4**, two N atoms from two SCN^−^ anions and two oxygen atoms from two dmso molecules form a square-planar environment for the Cu^2+^ ion. Due to the presence of two S(SCN) atoms at the apical positions of the copper ion coordination sphere (Cu...S 3.016 Å), the metal coordination number can be described as 4 + 2 (Figure 5). The Cu–N distances of 1.939(2) Å correspond to those observed in the copper(II) environment in Cu_3_(SCN)_4_(dmso)_2_ [9]. The Cu–O (dmso) 1.978(2) Å distances are typical of Cu^2+^—dmso bonds. Due to the bridging role of thiocyanate anions, infinite chains running along 100 crystallographic directions appear. These chains are connected only by non-valent interactions (Figure 6).

Compound **5** consists of Hdabco^+^ cations and SCN^−^ anions associated via N-H…N rather strong (N…N 2.763(2) Å) hydrogen bonds (Figure 7). (Hdabco)SCN moieties are bound via weak interactions only. The Hdabco moiety has an untwisted geometry. The HN-C and N-C distances are clearly different (Appendix A).

In compound **6**, both protonated nitrogen atoms of the H_2_dabco^2+^ di-cations are connected to the SCN^-^ anions via N-H…N hydrogen bonds (Figure 8a). The N…N distances 2.696(2) Å are even shorter than those in compound 5. (H_2_dabco)(SCN)_2_ moieties are bound via short S...S interactions of 3.46 Å [12] to form infinite chains (Figure 8b). The organic cation demonstrates untwisted geometry with HN-C distances similar to those of many known H_2_dabco salts.

In compound **7**, the water molecule plays out its coordination abilities to the fullest, as it is bonded to two (H_2_dabco)(SCN)_2_ moieties via N-H…O hydrogen bonds and to two CSN anions via O-H…N bonds. Due to the above bonds, infinite chains [(H_2_dabco)H_2_O (SCN)_2_)]_n_ running along 100 crystallographic directions appear (Figure 9a). These chains are interconnected via non-valent interactions only. The H_2_dabco^2+^ unit has an untwisted geometry with equal N-C and nearly equal C-C bond lengths (Appendix A).

### 2.3. Raman Spectra

First, it should be noted that the spectra of some of the studied compounds suffer from noticeable luminescence, resulting (as for compound **3**) the worst quality of the spectrum. The main feature of the Raman spectra of all the compounds studied is the presence of extremely strong peaks originating from the SCN^−^ group. Two other well-separated spectral regions are the area of the C-H and N-H modes and the region below 1500 cm^−1^. The peaks corresponding to the N–H and C–H modes split in all spectra.

Nakamoto [13] reports that ν_1_ frequency of SCN^−^ vibration appears below 2050 cm^−1^ in the case of M–NCS binding, around 2100 cm^−1^ for M–SCN bonding, and over 2100 cm^−1^ in the case of M–SCN–M bridging mode.

ν_1_Bonding type1.2103–2119Cu^+^–SCN–Cu^+^3.2090–2120Cu^2+^–NCS4.2095–2170Cu^2+^–NCS...Cu^2+^5.2048N-H...NCS6.2053N-H...NCS7.2078–2121O-H...NCS

The ν_1_ frequency in compounds **1**–**7** follows roughly the above-mentioned empirical rules. The lowest values were observed in the case of N-H...NCS coordination, and the highest were bridging thiocyanate anions.

A group of peaks observed in the Raman spectra of compounds **1**, **3,** and **5***–***7** almost coincide with those in the spectra of pure dabco [14] and [H_2_dabco]CuCl_3_ [15], since they originate from the organic part (Appendix A, Table 2).

The S-O stretching mode was found to be red shifted in 4 (line at 1000 cm^−1^), whereas a free isolated dmso molecule possesses a single adsorption line at 1053 cm^−1^ [16]. This shift is consistent with those found in a number of other dmso transition metal complexes [17]. Valence symmetric and asymmetric modes of the C-S bond were found in the spectrum of **7** at 716 cm^−1^ and 680 cm^−1^, respectively, and nearly coincide with those in the spectrum of pure dmso [17].

## 3. Materials and Methods

All chemicals were of commercial origin: 1,4-diazabicyclo [2.2.2] octane from Alfa Aesar, Ward Hill, Massachusetts, USA., 98%; dmso from Merck, Darmstadt, Germany, p.a.; DMF from Merck, >95%.; CuSCN from Thermo Scientific, Waltham, Massachusetts, USA, 96%; copper powder <425 μm, Sigma-Aldrich, Burlington, Massachusetts, USA, 99.5%; NH_4_SCN from Kefo, Ljubljana, Slovenia, >97%. All chemicals were used as received. The [Hdabco]Cl compound was prepared by passing dry HCl gas (generated by dropping H_2_SO_4_ on NaCl) through an ethanolic solution of dabco [18]. All experiments were carried out in the air.

### 3.1. Synthesis

#### 3.1.1. Syntheses of (Hdabco)[Cu_2_(NCS)_3_] (**1**) and [Cu(NCS)_2_(dmso)_2_] (**4**)

Copper powder (0.06 g, 1 mmol), NH_4_NCS (0.15 g, 2 mmol), dabco (0.11 g, 1 mmol), dmf/dmso (1:1) solvent mixture (10 mL) acidified with one drop (ca. 0.05 mL) of 10% HNO_3_ were heated to 60 °C and magnetically stirred for 30 min. Insignificant residues of unreacted reagents were filtered off. The colorless fine prismatic crystals of 1 were collected by filtration after 1 day, washed with isopropanol, and dried in the open air at r.t. The resulting red-brown solution was evaporated under magnetical stirring at 80 °C up to almost 3 mL and left in the open air. After a week, dark red well-defined needle crystals began to form in the solution, which turned into large light green rectangular crystals during further storage. Both types of crystals were tested by single-crystal X-ray analysis that affords to identify the known [Cu^I/II^_3_(NCS)_4_(dmso)_2_] (red) and new [Cu(NCS)_2_(dmso)_2_] (4, green) compounds. The yield of 1: 0.1 g, 47% (per copper). Anal. Calcd. for 1 (C9H13Cu2N5S3, Mr = 414.50): C, 26.08; H, 3.16; N, 16.90; S, 23.21; Cu, 30.66%. Found: C, 26.5; H, 2.8; N, 17.1; Cu, 32.0%.

#### 3.1.2. Synthesis of [Cu(Hdabco)_2_(NCS)_4_]∙2dmso (**3**)

Copper powder (0.03 g, 0.5 mmol), NH_4_NCS (0.38 g, 5 mmol), dabco (0.11 g, 1 mmol), and dmf/dmso (1:1) solvent mixture (20 mL) acidified with two drops (ca. 0.1 mL) of 10% HNO_3_ were heated to 90 °C and stirred magnetically for 2 h until the copper was almost completely dissolved. The resulting intense dark red solution was filtered. The green block crystals were collected by filtration after 5 days, washed with isopropanol, and dried in the open air at r.t. Yield: 0.5 g, 36.5% (per copper). Anal. Calcd. for C20H38CuN8O2S6 (Mr = 678.48): C, 22.06; H, 2.96; N, 30.88; S, 11.78; Cu, 11.67%. Found: C, 22.1; H, 3.0; N, 30.9; Cu, 11.7%.

#### 3.1.3. Synthesis of (H_2_dabco)[Cu(NCS)_3_] (**2**) Crystals Suitable for X-ray

CuSCN (0.06 g, 0.5 mmol), NH_4_NCS (0.08 g, 1 mmol), dabco (0.06 g, 0.5 mmol), dmso (7 mL) acidified with two drops (ca. 0.1 mL) of 10% HNO_3_ were heated to 60 °C and magnetically stirred for 20 min. Insignificant residues of unreacted reagents were filtered off. The colorless fine crystalline precipitate was collected by filtration after 1 day, washed with isopropanol, and dried in the open air at r.t. Single-crystal X-ray analysis carried out for the precipitate revealed a minimum of two compounds in the resulting mixture, namely, (Hdabco)[Cu_2_(NCS)_3_] (**1**) and (H_2_dabco)[Cu(NCS)_3_] (**2**). Since all crystals were small and similar, it was not possible to separate them in order to obtain the pure sample of 2 for further investigations.

#### 3.1.4. Synthesis of (Hdabco)(NCS) (**5**)

Two alternative methods were used for the synthesis of **5**.

(a)Dabco (0.11 g, 1 mmol) and NH_4_NCS (0.08 g, 1 mmol) were dissolved in methanol (20 mL). The resultant solution was stirred magnetically at room temperature for 15 min and left for slow evaporation in the open air until the total volume was about 2 mL. The colorless plate crystalline precipitate was collected by filtration and dried in the open air. Yield: 0.10 g, 67%. Anal. Calcd. for C7H13N3S (Mr = 171.26): C, 49.09; H, 7.65; N, 24.54; S, 18.72%. Found: C, 49.0; H, 8.0; N, 25.0%.(b)Dabco∙HCl (0.07 g, 0.5 mmol) and KNCS (0.05 g, 0.5 mmol) were dissolved in acetone (15 mL) and stirred magnetically at room temperature for 30 min. The white precipitate of KCl was filtered, and the solution was left for slow evaporation in the open air. The colorless plate crystals were collected by filtration and dried in the open air at r.t. Yield: 0.08 g, 88%. Anal. Found: C, 48.7; H, 7.1; N, 24.7%.

#### 3.1.5. Synthesis of (H_2_dabco)(NCS)_2_ (**6**)

Dabco (0.11 g, 1 mmol) and NH_4_NCS (0.16 g, 2 mmol) were dissolved in methanol (25 mL) at r.t. The resultant solution was stirred magnetically at room temperature for 15 min and left for slow evaporation in the open air until the total volume was about 2 mL. The colorless prismatic crystalline precipitate was collected by filtration and dried in the open air. Yield: 0.13 g, 55%. Anal. Calcd. for C8H14N4S2 (Mr = 230.35): C, 41.71; H, 6.13; N, 24.32; S, 27.84. Found: C, 42.0; H, 6.7; N, 24.4%.

### 3.2. Crystallography

Single-crystal X-ray data were collected on a Gemini A diffractometer equipped with an Atlas CCD detector, using graphite monochromated MoKα radiation. Data were processed using the CrysAlisPro software suite program package [19]. Analytical absorption corrections were applied to all data sets. All structures were solved using the dual-space algorithm of the SHELXT [20] program implemented in the Olex crystallographic software [21]. Structure refinement was performed with the SHELXL-2014 software [22]. Refinement of structure I converged at a relatively high (~0.05) R-factor value, and the highest peak on the residual electron density map exceeded ~5 times the value of the deepest hall. Careful inspection of the data using the TwinRotMat procedure implemented in Platon software revealed the presence of pseudo-merohedral twinning [23]. The twinning matrix was determined, and a new refinement using the HKLF4 data file resulted in a drastic decrease in the highest peak of the electron density and a decrease in the R-value. The positions of hydrogen atoms of amino groups and water molecules were found on a different Fourier map. Their positional and isotropic thermal parameters were freely refined, and other hydrogen atoms were placed in ideal positions and refined as riding atoms with relative isotropic displacement parameters. Figures were prepared using DIAMOND 4.6 [24] software. Topological analysis was performed using the TopCryst web service (http://topcryst.com, accessed on 20 March 2023). A summary of the crystallographic data and structure refinement are given in Table 1. CCDC 2252818 (1), 2252817 (2), 2252813 (3), 2252815 (4), 2252816 (5), 2252814 (6), 2252812 (7) contain the crystallographic data for this paper. These data can be obtained free of charge from The Cambridge Crystallographic Data Centre via www.ccdc.cam.ac.uk/data_request/cif, accessed on 20 March 2023.

### 3.3. Raman Spectroscopy

Raman spectra with a resolution of 0.5 cm^−1^ were recorded at room temperature on a Horiba Jobin Yvon LabRam-HR spectrometer equipped with an Olympus BXFM-ILHS microscope. Samples were excited by the 632.8 nm emission line of a He−Ne laser with a regulated power in the range 20−0.0020 mW, which provided 17−0.0017 mW focused on a 1 μm spot through a 50× microscope objective on the top surface of the sample. Before each measure, the spectrometer scale was additionally calibrated (a silicon polycrystalline plate, a characteristic band at 521 cm^−1^).

### 3.4. FTIR Spectroscopy

IR spectra were recorded on the Perkin-Elmer (Waltham, MA, USA) Frontier spectrometer.

### 3.5. Elemental Analysis

Sulfur, carbon, hydrogen, and nitrogen contents were determined using a CHNS elemental analyzer vario EL cube (Elementar) operating in the CHNS mode. The copper content was determined by complexometric titration.

## 4. Conclusions

Three new compounds based on copper thiocyanate and protonated dabco were obtained by the “direct synthesis” method. In the case of monoprotonated dabco, an anionic framework of {Cu_2_(SCN)_3_^−^}_n_ composition with infinite channels appears. Increasing the charge of the organic cation leads to the formation of an island-type structure containing a previously unknown [Cu(SCN)_3_]^2−^ anion. In the case of the Cu^2+^ derivative, a monoprotonated Hdabco moiety acts as a ligand bonded to the metal ion via a non-protonated nitrogen atom. A number of unknown by-products were separated and studied. The dependence of the formation of certain products on the synthesis conditions is not entirely clear and will be the subject of further experiments.

## Data Availability

Not applicable.

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
