# Peer review of "Cation Charge as a Tool to Change Dimensionality in Organic–Inorganic Hybrids Based on Copper Thiocyanate Templated by 1,4-Diazabicyclo[2.2.2]octane"

_molecules, 2023, doi:10.3390/molecules28083608_

Round 1
Reviewer 1 Report
The manuscript reported the syntheses and characterizations of a series of compounds based on copper/thiocyanate/1,4-diazabicyclo[2.2.2]octane(dabco). The crystal structures of all these compounds were also determined by single-crystal X-ray diffraction method. The quality of structure data was pretty good. Although the similar compounds were studied and revealed by the authors and other people in this field in literatures, the current research reported in this manuscript is quite interesting and a good addition to this field with thorough characterizations. I think that the manuscript is suitable for publication in Molecules with minor revisions that should be considered to correct by the authors.
1. The reference labels should keep consistent. Most of time authors used Roman numerals, while in one case the Arabic numeral was used (line 250). It would be better to use Arabic numerals to label the references rather than to use Roman numerals.
2. Although the quality of reported structures seems pretty good from the table 1 in the manuscript, it is a little bit not convenient for me to fully examine the structure refinement without cif files and the related checkcif reports. These documents should be included with the manuscript.
3. There may be a typo in line 85: the “… structure I …” should be changed to “structure 1” to keep in consistent with the label in Table 1.
4. There may be a typo in line 131 and line 172: the “… [Error! Bookmark not defined., Error! Bookmark not defined.]…” should be deleted.
Author Response
Dear Editor and Reviewers,
Thank you for your comments and recommendations regarding our Manuscript “Cation charge as a tool to change dimensionality in organic-inorganic hybrids based on copper thiocyanate templated by 1,4-diazabicyclo[2.2.2]octane”.
All Manuscript was re-formated accordingly to Journal requirements. Description of crystal structure of compound 4 was improved, and two additional Figures regarding this structure were added. Also we expanded description of syntheses basing on our results obtained just last weeks.
Reviewer 1
- The reference labels should keep consistent. Most of time authors used Roman numerals, while in one case the Arabic numeral was used (line 250). It would be better to use Arabic numerals to label the references rather than to use Roman numerals.
All references are checked and corrected.
- Although the quality of reported structures seems pretty good from the table 1 in the manuscript, it is a little bit not convenient for me to fully examine the structure refinement without cif files and the related checkcif reports. These documents should be included with the manuscript.
All cif-files and checkcif files were added to Submission.
- There may be a typo in line 85: the “… structure I …” should be changed to “structure 1” to keep in consistent with the label in Table 1.
Corrected.
- There may be a typo in line 131 and line 172: the “… [Error! Bookmark not defined., Error! Bookmark not defined.]…” should be deleted.
Corrected.
Reviewer 2 Report
Good article, the relevance of the subject is related, among other things, to the possibility of obtaining in such systems MOF.
However, there are a few comments to the text of the paper.
1. Please to the authors to make references to literature and citation list according to the template of the journal.
2. In the experimental part, indicate how compounds (1) and (2) were obtained and separated. Are they formed as a mixture, or do they undergo a transformation of one into the other? It is also worth giving the elemental analysis data for both samples.
3. For copper coordination compounds obtained, XRD data should be given to determine if the samples are single-phase. In addition, the IR spectra should be very informative to determine the type of coordination of the ligand and confirm the presence of hydrogen bonds in the system.
4. The scheme of synthesis of the target compounds should be given considering the requirements of the journal - using graphical editors, for example, Chem Draw.
After the necessary corrections, the article can be published in Molecules.
Author Response
Dear Editor and Reviewers,
Thank you for your comments and recommendations regarding our Manuscript “Cation charge as a tool to change dimensionality in organic-inorganic hybrids based on copper thiocyanate templated by 1,4-diazabicyclo[2.2.2]octane”.
All Manuscript was re-formated accordingly to Journal requirements. Description of crystal structure of compound 4 was improved, and two additional Figures regarding this structure were added. Also we expanded description of syntheses basing on our results obtained just last weeks.
Reviewer 2
- Please to the authors to make references to literature and citation list according to the template of the journal.
All Manuscript was re-formated accordingly to Journal requirements.
- In the experimental part, indicate how compounds (1) and (2) were obtained and separated. Are they formed as a mixture, or do they undergo a transformation of one into the other? It is also worth giving the elemental analysis data for both samples.
Additional information on syntheses was added to Experimental and Results parts.
- For copper coordination compounds obtained, XRD data should be given to determine if the samples are single-phase. In addition, the IR spectra should be very informative to determine the type of coordination of the ligand and confirm the presence of hydrogen bonds in the system.
The IR spectra for compounds 1, 3 and 4 were measured and added to Supp. Materials.
All multy-phase products were carefully analyzed, checking at least 10 different crystal with different morphology from each case by X-ray and Raman techniques.
- The scheme of synthesis of the target compounds should be given considering the requirements of the journal - using graphical editors, for example, Chem Draw.
Respective Schemes were added to Manuscript.
Round 2
Reviewer 2 Report
The authors have done considerable work to improve the manuscript text. All comments have been taken into account.
The article can be published in its current form.